# Study of Thermodynamic Modeling of Isothermal and Isobaric Binary Mixtures in Vapor-Liquid Equilibrium (VLE) of Tetrahydrofuran with Benzene (303.15 K) Cyclohexane (333.15 K), Methanol (103 kPa), and Ethanol (100 kPa)

Leonardo Steyman Reyes Fernández , Eliseo Amado-Gonzaléz * and Erik Germán Yanza Hurtado

Facultad de Ciencias Básicas, Universidad de Pamplona, Pamplona 543057, Colombia;
leonardo.reyes@unipamplona.edu.co (L.S.R.F.); egeyanza@unipamplona.edu.co (E.G.Y.H.)
* Correspondence: eamado@unipamplona.edu.co; Tel.: +57-3-114-621-948

**Abstract:** Tetrahydrofuran (THF) is an aprotic solvent with multiple applications in diverse areas of chemical, petrochemical, and pharmaceutical industries with an important impact in chemical waste liquid with other solvents. In this work, 51 available VLE data, for isothermal binary mixtures of THF(1) + Benzene(2) and THF(1) + Cyclohexane(2) at 303.15 and 333.15 K, respectively, and isobaric THF(1) + Methanol(2) at 103 kPa and THF(1) + Ethanol(2) at 100 kPa were used in the development of the activity coefficient models. The quality of experimental data was checked using the Herington test. VLE binary data was correlated with models Wilson, NRTL UNIQUAC, and UNIFAC to obtain binary parameters and activity coefficients. The best thermodynamic consistency when conducting the Herington test for the VLE data was found for the THF(1) +Cyclohexane(2) isothermal system and THF(1) + Ethanol(2) isobaric system. The UNIQUAC model for isothermal systems THF(1) + Benzene(2) and THF(1) + Cyclohexane(2), the NRTL model for the isobaric system THF(1) + Methanol(2), and the UNIQUAC model for THF(1) + Ethanol(2) perform better than the other models.

**Keywords:** vapor liquid equilibrium (VLE); THF; Benzene; cyclohexane; methanol; ethanol RMSD; thermodynamic consistency

## 1. Introduction

Tetrahydrofuran (THF) is a five-member cyclic ether, widely used as a chemical intermediate in the production of polymers, such as polytetrahydrofuran (PolyTHF), has a great utility for the production of elastic fabrics, synthetic leather, clothing and apparel, compression garments, and household furniture [1]. While THF is used as a solvent in many useful chemical processes, manufacturing activities and active additives in the synthesis of pharmaceutical products, it is also found with other solvents, such as cyclohexane in chemical waste liquids [2]. The mixture of THF and cyclohexane presents an azeotrope at 338.74 K with the composition of THF at 93% (wt %) with an almost impossible separation by conventional distillation [3]. On the other hand, in polymerization reactions, THF is soluble in all proportions with alcohols, phenols, and all common solvents [4]. The separation of azeotropic multicomponent mixtures, such as THF, methanol and water, provide considerable potential for the combination of pervaporation and distillation processes for THF recovery with reliable benefits [5].

In the study of preferential interaction of polymers in mixed solvents, the binary mixtures of THF with aromatic hydrocarbons showed important changes at high THF concentrations [6]. Additionally, THF was used for the selective protonation of aromatic hydrocarbons with high selectivity in moderate to good yields [7] and in electro-reduction mechanism of aromatic hydrocarbons [8].

From the economical point and green chemistry viewpoint, the growing demand for THF from emerging markets is an important factor that is expected to provide opportunities for revenue growth for major players operating in the global THF market in the midst of the COVID-19 crisis [9]. In developing countries, such as Colombia, the imports of THF are around US $8,87,436 against exports of US $204,094 in the same period [10].

On the other hand, the problem of the thermodynamics of mixtures is of great interest, as evidenced by innumerable applications, also in the industrial field, which this research topic discusses. In mixtures where azeotropic systems are present, activity coefficients data are of great utility for the design of efficient separation and purification processes [11]. The objective of this work is based on the evaluation of activity coefficients and the mathematical modeling of vapor-liquid equilibrium (VLE) binary mixtures of THF/organic mixtures for the calculation of activity coefficients from literature data, using activity coefficient models such as local composition models (Wilson and NRTL) and a local distribution model (UNIQUAC).

## 2. Methodology

The interaction parameter's optimization for each model calculations were developed using the GRC resolution method software Microsoft Excel® complement.

### 2.1. Thermodynamic Databases

Experimental data used for this study were collected from thermodynamic databases, Dortmund data bank (DDB) and Korea data bank (KDB), and literature. The data were classified into two types: isothermal data at two different temperatures (303.15, 333.15) and two isothermal data at the pressure of 103 and 100 kPa (Table 1).

**Table 1.** Binary Vapor-Liquid Equilibrium Data.

| Binary System | Type | Reference |
|---|---|---|
| THF(1) + Benzene(2) | isothermal | [12] |
| THF(1) + Cyclohexane(2) | Isothermal | [13] |
| THF(1) + Methanol(2) | Isobaric | [14] |
| THF(1) + Ethanol(2) | Isobaric | [15] |

### 2.2. Theoretical Bases

The activity coefficients $\gamma_i$ were calculated by Equation (1)

$$y_i \Phi_i P = x_i \gamma_i P^{sat} \tag{1}$$

where $y_i$, $x_i$, $\Phi_i$, $\gamma_i$ and $P$ refer to the vapor phase composition, liquid phase composition, fugacity coefficient, activity coefficient, and equilibrium pressure, respectively. Since the pressure of the collected experimental data is less than 1, the gas phase can be assumed to be ideal behavior, then $i = 1$ [16].

#### 2.2.1. Wilson Model

For phase equilibrium calculations, activity coefficients are used to account within a liquid solution for local compositions that, in turn, differ from the overall composition of the mixture. In 1964, G.M. Wilson published a solution behavior model known as the Wilson model [17].

$$ln(\gamma_1) = -ln(x_1 + \Lambda_{12}x_2) + x_2\left(\frac{\Lambda_{12}}{x_1 + \Lambda_{12}x_2} - \frac{\Lambda_{21}}{x_2 + \Lambda_{21}x_1}\right) \tag{2}$$

$$ln(\gamma_2) = -ln(x_2 + \Lambda_{21}x_1) + x_1\left(\frac{\Lambda_{21}}{x_2 + \Lambda_{21}x_1} - \frac{\Lambda_{12}}{x_1 + \Lambda_{12}x_2}\right) \tag{3}$$

where $\Lambda_{ij}$ ($\Lambda_{21}$, $\Lambda_{12}$) is the adjustable parameter of the Wilson model.

### 2.2.2. Non-Random Two Liquids (NRTL)

The NRTL equation contains 3 parameters for the binary system [18].

$$ln\gamma_1 = x_2^2\left[\tau_{21}\left(\frac{G_{21}}{x_1 + x_2 G_{21}}\right)^2 + \frac{G_{12}\tau_{12}}{(x_2 + x_1 G_{12})^2}\right] \tag{4}$$

$$ln\gamma_2 = x_1^2\left[\tau_{12}\left(\frac{G_{12}}{x_2 + x_1 G_{12}}\right)^2 + \frac{G_{21} * \tau_{21}}{(x_1 + x_2 G_{21})^2}\right] \tag{5}$$

### 2.2.3. Universal Quasi-Chemical Model (UNIQUAC)

Even the mathematical expression of the UNIQUAC model is considered more complex than that of the NRTL model, and it is more commonly used in chemical engineering [19]. One of its advantages is that it has fewer adjustable parameters, two instead of three, which are less temperature dependent and can be applied to systems with larger size differences.

$$ln\gamma_i = ln\,\gamma_i^C + ln\,\gamma_i^R \tag{6}$$

$$ln\gamma_i^C = 1 - J_i + lnJ_i - 5q_i\left(1 - \frac{J_i}{L_i} + ln\frac{J_i}{L_i}\right) \tag{7}$$

$$ln\gamma_i^R = q_i(1 - lns_i - \sum_j \frac{\theta_j \tau_{ki}}{s_j}) \tag{8}$$

$$\tau_{ji} = \exp\left(-\frac{u_{ji} - u_{ii}}{RT}\right) \tag{9}$$

$$J_i = \frac{r_i}{\sum_j r_j x_j} \tag{10}$$

$$L_i = \frac{q_i}{\sum_j q_j x_j} \tag{11}$$

$$s_i = \sum_k \theta_k \tau_{ki} \tag{12}$$

$$\theta_i = \frac{x_i q_i}{\sum_j q_j x_j} \tag{13}$$

$$r_i = \sum_k v_k^{(i)} R_k \tag{14}$$

$$q_i = \sum_k v_k^{(i)} Q_k \tag{15}$$

In these equations, $r_i$ is a parameter representing a relative molecular volume and $q_i$ is a parameter representing a relative molecular surface area, each of which is given by the sum of $R_k$ and $Q_k$ parameters of functional groups comprising the component and listed Table 2.

**Table 2.** Structural parameters for the UNIQUAC equation.

| Component | $r_i$ | $q_i$ | Reference |
|---|---|---|---|
| Tetrahydrofuran | 2.9415 | 2.72 | [20] |
| Cyclohexane | 4.0464 | 3.24 | [20] |
| Benzene | 3.1878 | 2.4 | [20] |
| Methanol | 1.43 | 0.96 | [20] |
| Ethanol | 2.588 | 0.92 | [20] |

### 2.2.4. UNIQUAC Functional-Group Activity Coefficients (UNIFAC)

In the UNIFAC activity coefficient model, the fugacity of the component *i* in mixtures consists of two parts, the combined term and the residual term [21].

$$ln\gamma_i^C = ln\frac{\Phi_i}{x_i} + \frac{z}{2}q_i ln\frac{\theta_i}{\Phi_i} + l_i - \frac{\Phi_i}{x_i}\sum_j x_j l_j \tag{16}$$

$$ln\gamma_i^R = \sum_k v_k^{(i)}(ln\Gamma_k - ln\Gamma_k^{(i)}) \tag{17}$$

$$l_i = \frac{z}{2}(r_i - q_i) - (r_i - 1) \tag{18}$$

where $z = 10$

$$\theta_i = \frac{x_i q_i}{\sum_j x_j q_j} \tag{19}$$

$$\Phi_i = \frac{x_i r_i}{\sum_j x_j r_j} \tag{20}$$

The Equations (23) and (24) correspond to the area fraction and segment fraction of the component *I* (Table 3).

$$r_i = \sum_k v_k^{(i)} R_k \tag{21}$$

$$q_i = \sum_k v_k^{(i)} Q_k \tag{22}$$

$$ln\Gamma_k = Q_k[1 - (ln(\sum_m \theta_m \Psi_{mk}) - \sum_m \frac{\theta_m \Psi_{km}}{\theta_n \Psi_{nm}}) \tag{23}$$

$$\theta_m = \frac{X_m Q_m}{\sum_n X_n Q_n} \tag{24}$$

$$X_m = \frac{\sum_j v_m^{(j)} x_j}{\sum_n \sum_j v_n^{(j)} x_j} \tag{25}$$

Equations (25) and (26) correspond and are the volume and surface of the group.

$$\Psi_{mk} = \exp\left(\frac{-a_{mk}}{T}\right) \tag{26}$$

**Table 3.** Structural parameters for the UNIFAC equation.

| Component | *r* | *q* | Reference |
|---|---|---|---|
| Tetrahydrofuran | 2.9415 | 2.72 | [18] |
| Cyclohexane | 4.0464 | 3.24 | [18] |
| Benzene | 3.1878 | 2.4 | [18] |
| Methanol | 1.43 | 1.432 | [18] |
| Ethanol | 2.588 | 2.588 | [18] |

In Figures 1–4, the comparison between experimental data of THF(1) + Benzene(2), THF(1) + Cyclohexane(2), THF(1) + Methanol(2), and THF(1) + Ethanol correlated are presented.

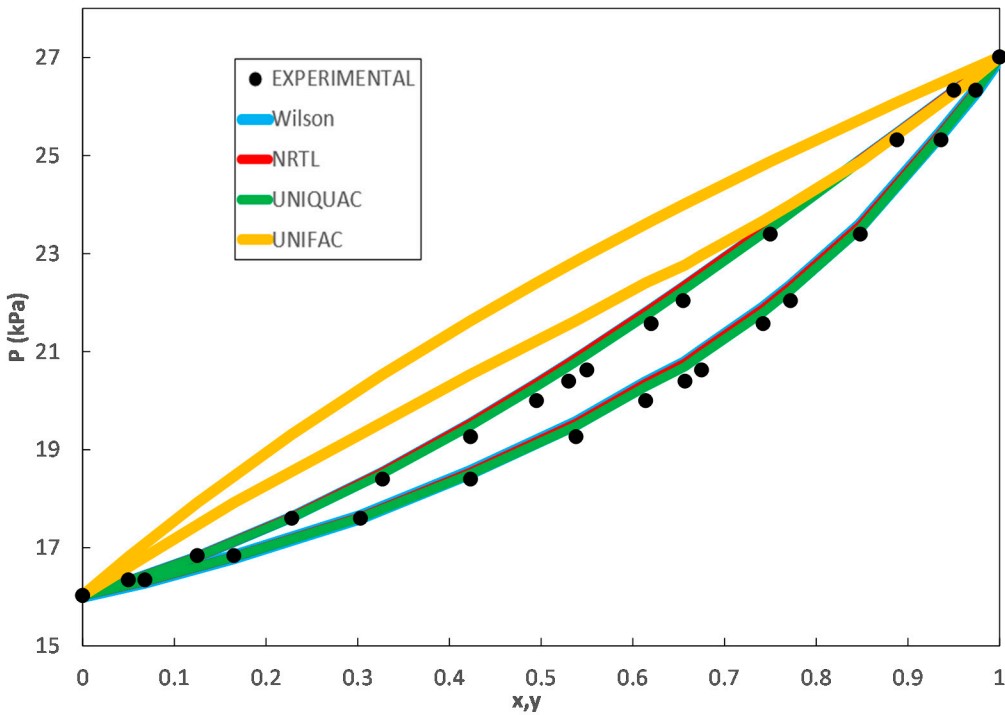

**Figure 1.** Experimental [12] and predicted P-xy diagram for THF(1) + Benzene(2) at 303.15 K using Wilson, NRTL, UNIFAC, and UNIQUAC as predictive models.

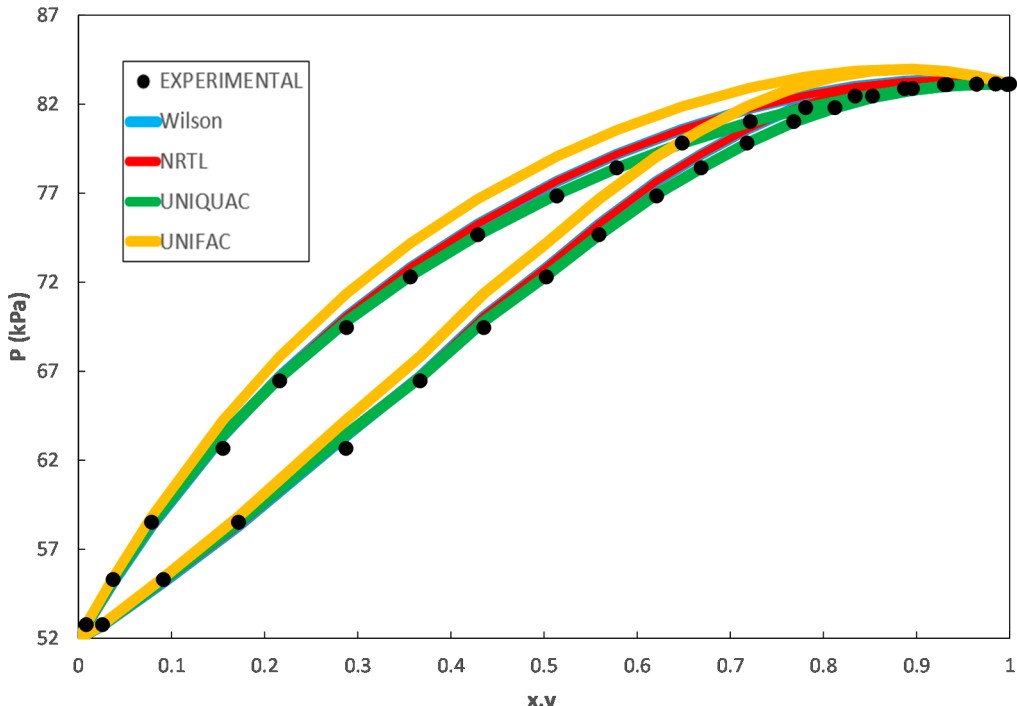

**Figure 2.** Experimental [13] and predicted P-xy diagram for THF(1) + Cyclohexane(2) at 333.15 K using Wilson, NRTL, UNIFAC, and UNIQUAC as predictive models.

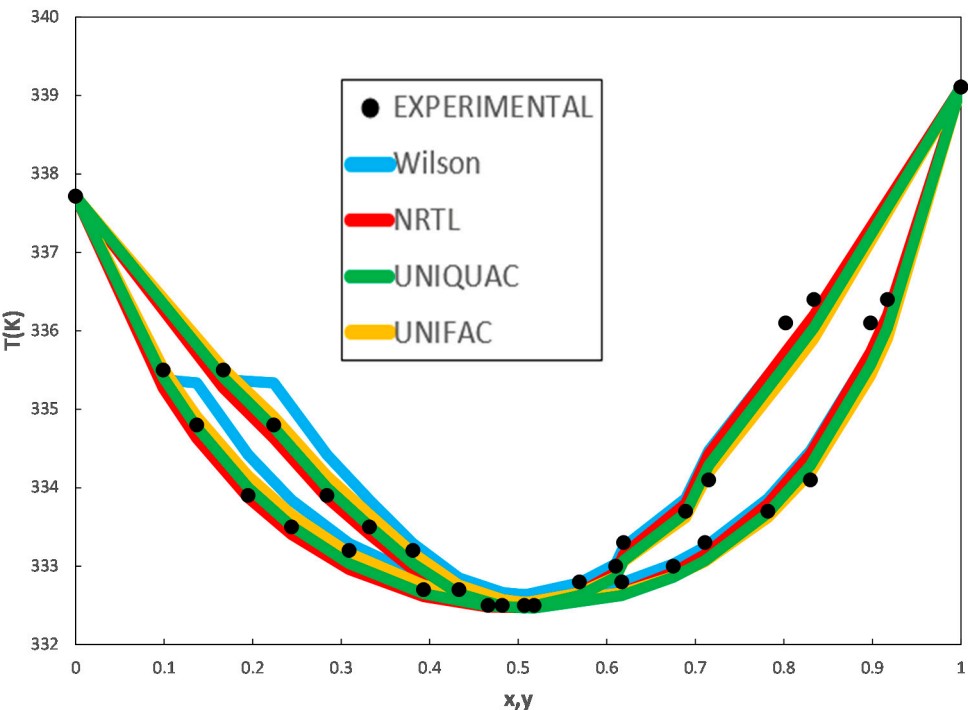

**Figure 3.** Experimental [14] and predicted T-xy diagram of THF(1) + Methanol(2) at 103 kPa using Wilson, NRTL, UNIFAC, and UNIQUAC as predictive models.

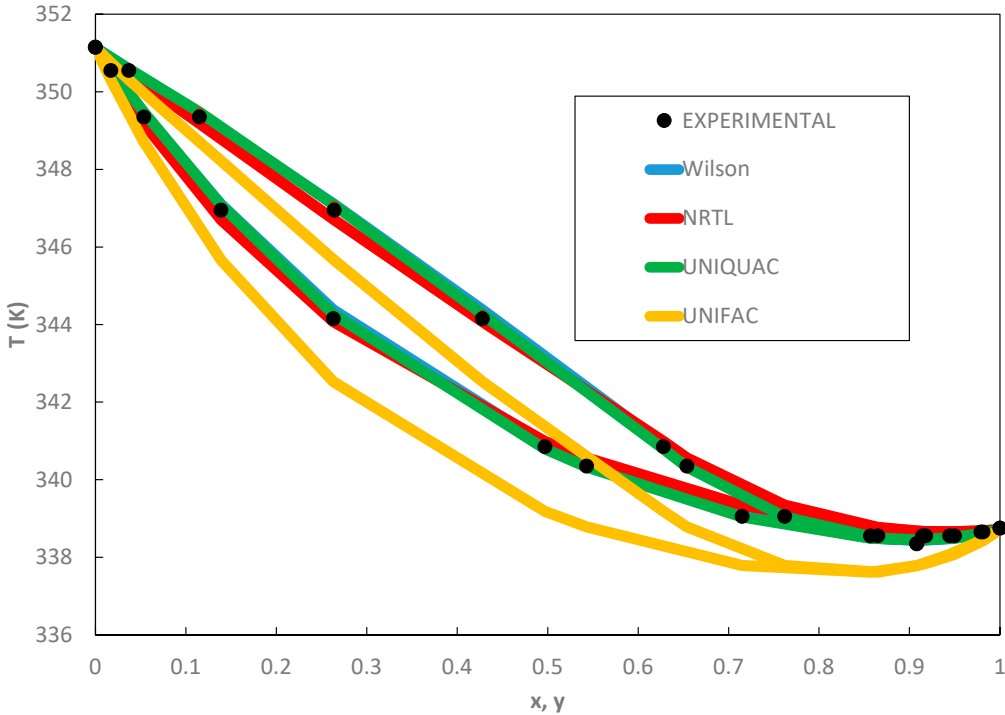

**Figure 4.** Experimental [15] and predicted T-xy diagram of THF(1) + Ethanol(2) at 100 kPa using Wilson, NRTL, UNIFAC, and UNIQUAC as predictive models.

### 2.3. Thermodynamic Consistency of the Experimental Data

If a data can be predicted by different measurements and mathematical relationships, then consistency can be claimed if the predicted and experimental values match to within experimental uncertainty [22]. In the current study, the isobaric binary data systems THF(1) + Methanol(2), and THF(1) + Ethanol(2) and the isothermal binary data systems THF(1)

+ Benzene(2) and THF(1) + Cyclohexane(2) are fitted to Wilson, NRTL, UNIQUAC and UNIFAC models. VLE data literature was used to calculate the interaction parameters. The quality of the experimental data was checked using the consistency the Herington thermodynamic test [23] used in the NIST ThermoData Engine [24].

Herington's method based on Gibbs Duhem's theory (Equation (27)) calculates the area under the two curves.

$$\int_0^1 ln\left(\frac{\gamma_1}{\gamma_2}\right)dx_1 = -\int_{P_2^{vap}}^{P_1^{Vap}} \frac{\widetilde{V}^{exp}}{RT}dP + \int_{T_2^{sat}}^{T_1^{sat}} \frac{\hat{H}^{exp}}{RT^2}dT \tag{27}$$

By definition, if data can be predicted by different measurements and mathematical relationships, then consistency can be claimed if the predicted and experimental values match to within experimental uncertainty [22].

When the VLE measurements are made under constant temperature, the Equation (27) simplifies to:

$$\int_0^1 ln\left(\frac{\gamma_1}{\gamma_2}\right)dx_1 = -\int_{P_2^{vap}}^{P_1^{Vap}} \frac{\widetilde{V}^{exp}}{RT}dP \tag{28}$$

where the volume change may be considered negligible for all systems. Under these conditions, the right-hand side of Equation (28) is almost zero.

When the data is represented in the form of $ln(\gamma_1/\gamma_2)$ versus $x_1$, as shown in Figure 5, the areas above (a) and below (b) the $x_1$ axis must be equal.

$$A = 100\left|\int_0^1 ln\left(\frac{\gamma_1}{\gamma_2}\right)dx_1\right| < 3 \tag{29}$$

$$D = 100\left[\frac{|A|}{a + |b|}\right] < 10 \tag{30}$$

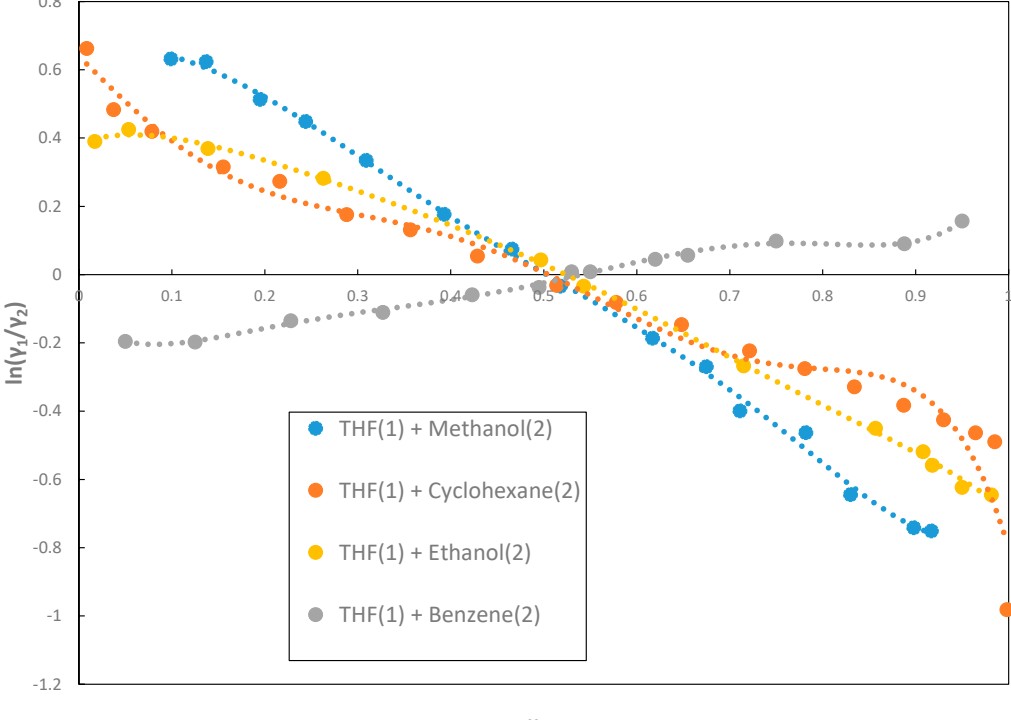

**Figure 5.** Herington test for isobaric and isothermal systems between THF(1)+ Methanol(2) at 103 kPa [12], THF(1) + Cyclohexane(2) at 333.15 K [13], THF(1) + Ethanol(2) at 100 kPa [14], and THF(1) + Benzene(2) at 303.15 K [15].

For isobaric VLE measurements, Equation (27) simplifies to:

$$\int_0^1 ln\left(\frac{\gamma_1}{\gamma_2}\right)dx_1 = \int_{T_2^{sat}}^{T_1^{sat}} \frac{\hat{H}^{exp}}{RT^2}dT \tag{31}$$

$$J = 150\frac{T_{max} - T_{min}}{T_{min}} \tag{32}$$

$$|\mathbf{D} - \mathbf{J}| < 10 \tag{33}$$

where, $T_{max}$ and $T_{min}$ represent the maximum and minimum temperatures in this study [25]. On Tables 4 and 5, the values of the consistence using the Herington test for isothermal and isobaric systems are presented.

**Table 4.** The results of thermodynamic consistency test using the Herington test for the isothermal systems.

| System | D | Consistency |
|---|---|---|
| THF(1) + Benzene(2) | 20.62 | (−) |
| THF(1) + cyclohexane(2) | 3.32 | (+) |

**Table 5.** The results of thermodynamic consistency when using the Herington test for the isobaric system.

| System | D \|D−J\| | Consistency |
|---|---|---|
| THF(1) + Methanol(2) | 3.440 1.6806 | (+) |
| THF(1) + Ethanol(2) | 7.6011 2.1925 | (+) |

### 2.4. Correlation of VLE Data

The binary parameters in the five coefficients equations were estimated based on the objective function OF in terms of the calculated and experimental activity coefficient.

$$OF = \sum_{i=i}^{N}\sum_{j=1}^{2}\left[\left(\frac{\gamma_j^{exp} - \gamma_j^{cal}}{\gamma_j^{exp}}\right)^2\right] \tag{34}$$

where *i* represents the amount of data from 1 to *N* and *j* denotes the number of components in the system.

The obtained interaction parameters of Wilson, NRTL, and UNIQUAC models with the objective function (OF) are listed in Table 6.

**Table 6.** The results of binary parameters using OF

| Model | *A*ij (J/mol) | *A*ji (J/mol) | αij |
|---|---|---|---|
| | THF(1) + Benzene(2) | | |
| Wilson | 0.9699 | 1.2618 | |
| NRTL | −938.057 | 386.177 | 0.3 |
| UNIQUAC | −8.2597 | −50.7358 | |
| | THF(1) + Cyclohexane(2) | | |
| Wilson | 0.7778 | 0.7197 | |
| NRTL | 954.65 | 628.29 | 0.3 |
| UNIQUAC | −99.129 | 174.79 | |
| | THF(1) + Methanol(2) | | |
| Wilson | 0.6960 | 0.5356 | |
| NRTLUNIQUAC | 1154.3955340 | 1423.46−135.25 | 0.3 |
| | THF(1) + Ethanol(2) | | |
| Wilson | 1.1022 | 0.4420 | |
| NRTL | 767.15 | 933.81 | 0.3 |
| UNIQUAC | 579.22 | −232.99 | |

The root-mean-square deviations (RMSD) were employed to evaluate the difference between the experimental and calculated results. The $RMSDy_i$ and $RMSDT_i$ for the isobaric systems and $RSMDP_i$ for isothermal systems are listed in Tables 7 and 8.

$$RMSDP_i = \sqrt{\sum_{i=1}^{N} \frac{(P_i^{exp} - P_i^{cal})^2}{N}} \tag{35}$$

$$RMSDT_i = \sqrt{\sum_{i=1}^{N} \frac{(T_i^{exp} - T_i^{cal})^2}{N}} \tag{36}$$

$$RMSDy_i = \sqrt{\sum_{i=1}^{N} \frac{(y_i^{exp} - y_i^{cal})^2}{N}} \tag{37}$$

**Table 7.** RMSD for the isothermal systems.

| Model | $RMSDP_i$ | $RMSDy_i$ |
|---|---|---|
| THF(1) + Benzene(2) | | |
| Wilson | 0.2335 | 0.0112 |
| NRTL | 0.2334 | 0.012 |
| UNIQUAC | 0.1854 | 0.0121 |
| UNIFAC | 1.807 | 0.0633 |
| THF(1) + Cyclohexane(2) | | |
| Wilson | 0.5227 | 0.0076 |
| NRTL | 0.5257 | 0.0077 |
| UNIQUAC | 0.2048 | 0.0076 |
| UNIFAC | 1.4562 | 0.0363 |

**Table 8.** RMSD for the isobaric system.

| Model | $RMSDT_i$ | $RMSDy_i$ |
|---|---|---|
| THF(1) + Methanol(2) | | |
| Wilson | 0.2739 | 0.0088 |
| NRTL | 0.1823 | 0.0077 |
| UNIQUAC | 0.2482 | 0.0089 |
| UNIFAC | 0.1998 | 0.0109 |
| THF(1) + Ethanol(2) | | |
| Wilson | 0.1011 | 0.005 |
| NRTL | 0.1894 | 0.0069 |
| UNIQUAC | 0.081 | 0.0053 |
| UNIFAC | 1.0637 | 0.0302 |

From Table 7, the UNIQUAC model for isothermal systems THF(1) + Benzene(2) and THF(1) + Cyclohexane(2) performs better than NRTL, UNIFAC, and Wilson models. From Table 8, the NRTL model for the isobaric system THF(1) + Methanol(2) and UNIQUAC model THF(1) + Ethanol(2) fitted the experimental data better than the others.

### 3. Conclusions

The experimental isothermal VLE data [12,13] and the predicted data for the binary system of THF(1) + Benzene(2) and THF(1) + Cyclohexane(2) at 303.15 and 333.15 K, respectively, isobaric VLE Data [14,15], and the predicted THF(1) + Methanol(2) and THF(1) + Ethanol(2) at 103 kPa were correlated by Wilson, NRTL, UNIQUAC, and UNIFAC models. Using the Herington test, the best thermodynamic consistency for the VLE data was found for the THF(1) + Cyclohexane(2) isothermal system and the THF(1) + Ethanol(2) isobaric system. The $RMSDP_i$ and $RMSDy_i$ show that the UNIQUAC model for isothermal systems THF(1) + Benzene(2) and THF(1) + Cyclohexane(2) and the NRTL model for the isobaric

systems THF(1) + Methanol(2) and UNIQUAC model for THF(1) + Ethanol(2) perform better than the other models. Probably, as previously proposed, the non-equilibrium state between phases in the VLE is focused around the variations of vapor flow quality, entropy generation, and exergy variation [26]. The analysis of the phase equilibrium in magnetorheological fluids could be a second research opportunity for non-equilibrium systems [27,28].

**Author Contributions:** Data curation, L.S.R.F. and E.G.Y.H. Formal analysis, E.A.-G., L.S.R.F. and E.G.Y.H.; Investigation, E.A.-G., L.S.R.F. and E.G.Y.H.; Methodology, E.A.-G.; Software, L.S.R.F.; Supervision, E.A.-G. and E.G.Y.H.; Visualization, E.G.Y.H.; Writing—original draft, L.S.R.F.; Writing—review & editing, E.A.-G. All authors have read and agreed to the published version of the manuscript.

**Funding:** This research received no external funding.

**Data Availability Statement:** Dortmund Data Bank Software Package (DDBSP): http://www.ddbst.de. KDB (Korea Thermophysical Properties Data Bank): http://www.cheric.org/research/kdb.

**Conflicts of Interest:** The authors declare no conflict.

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
