# Peer review of "Study of Thermodynamic Modeling of Isothermal and Isobaric Binary Mixtures in Vapor-Liquid Equilibrium (VLE) of Tetrahydrofuran with Benzene (303.15 K) Cyclohexane (333.15 K), Methanol (103 kPa), and Ethanol (100 kPa)"

_2673-7264, doi:10.3390/thermo1030019_

Round 1
Reviewer 1 Report
*) There is a need to provide more comments and details to the Figures present in the paper.
*) At the same time, it would be desirable to make the captions of the figures self-explanatory.
*) Not all mathematical formulas in the text are numbered. Please remedy.
*) Please read the text carefully to correct any punctuation errors.
*) Please read the text carefully to eliminate any typos.
*) In (1) are variables vectors? It is not clear why the formula is shown in bold.
*) Are the adjustable parameter of the Margules model the same as those present in the Wilson model? Please specify why the same symbols appear.
*) The authors wrote: "The mathematical expression of the UNIQUAC model is more complex than that of the NRTL model, but is more commonly used in chemical engineering. "Please provide some literature references.
*) What do the authors mean by "Consistency of the Experimental Data"? Please specify better in the text.
*) In (31), please define in detail the function to be integrated in order to clearly highlight that it satisfies the integrability requirements.
*) In the conclusions it would be desirable to outline at least one line of future research.
*) The problem of the thermodynamics of mixtures is of great interest as evidenced by the innumerable applications, also in the industrial field, that this research topic involves.
*) In particular, some types of mixtures require that the problem is not framed in the context of classical thermodynamics but rather in the extended irreversible thermodynamics. This is the case of magnetorheological fluids and, in particular, of certain types of evolutionary models of fludi mixture. Without prejudice to the fact that this topic is beyond the objectives of the work presented by the authors, I recommend inserting at least one sentence in the text that highlights this problem by putting the following relevant works in the bibliography:
doi: 10.1016/j.ijnonlinmec.2019.103288
doi: 10.1007/s001610200085
Author Response
*) There is a need to provide more comments and details to the Figures present in the paper.
ANSWER: Thanks for your observation. The corrections were done.
Figure 1.
Figure 2.
Figure 3.
Figure 4.
Figure 5.
*) At the same time, it would be desirable to make the captions of the figures self-explanatory
ANSWER: Thanks for your observation. The corrections were done.
Figure 1
Figure 2
Figure 3
Figure 4.
Figure 5.
*) Not all mathematical formulas in the text are numbered. Please remedy.
ANSWER: Thanks for your observation. The correction was done.
*) Please read the text carefully to correct any punctuation errors.
ANSWER: Thanks for your observation. The correction was done.
*) Please read the text carefully to eliminate any typos.
ANSWER: Thanks for your observation. The correction was done.
*) In (1) are variables vectors? It is not clear why the formula is shown in bold.
ANSWER: Thanks for your observation. The correction was done.
*) Are the adjustable parameter of the Margules model the same as those present in the Wilson model? Please specify why the same symbols appear.
ANSWER: Thanks for your observation. The correction was done. Of course the parameters of each model are different. In our case we decided not to use Margules and van Laar models for this paper
*) The authors wrote: "The mathematical expression of the UNIQUAC model is more complex than that of the NRTL model, but is more commonly used in chemical engineering. "Please provide some literature references.
Answer: Thanks for your observation. The correction was done: 19. Abrams, D. S.; Prausnitz, J. M. Statistical Thermodynamics of Liquid Mixtures: A New Expression for the Excess Gibbs Energy of Partly or Completely Miscible Systems. AIChE J. 1975, 21, 116–128.
*) What do the authors mean by "Consistency of the Experimental Data"? Please specify better in the text.
Answer: Thanks for your observation. The correction was done.
By definition if a data can be predicted by different measurements and the mathematical relationships, then consistency can be claimed if the predicted and experimental values match to within experimental uncertainty [19].
*) In (31), please define in detail the function to be integrated in order to clearly highlight that it satisfies the integrability requirements.
Answer: Thanks for your observation.
By definition if a data can be predicted by different measurements and the mathematical relationships, then consistency can be claimed if the predicted and experimental values match to within experimental uncertainty [22].
When the VLE measurements are made under constant temperature and equation 27 simplifies to
∫_0^1▒〖ln(γ_1/γ_2 )dx_1=-∫_(P_2^vap)^(P_1^Vap)▒〖V ̃^exp/RT dP〗〗 (28)
Where, the volume change may be considered negligible for all systems. Under these conditions, the right-hand side of equation 28 is almost zero.
When the data is represented in the form of ln(γ1/γ2) versus x1, as shown in Figure 5, the areas above (a) and below (b) the x1 axis must be equal.
A=100|∫_0^1▒〖ln(γ_1/γ_2 )dx_1 〗|<3 (29)
D=100[|A|/(a+|b| )]<10 (30)
For isobaric VLE measurements equation 27 simplifies to
∫_0^1▒〖ln(γ_1/γ_2 )dx_1=∫_(T_2^sat)^(T_1^sat)▒〖H ̂^exp/(RT^2 ) dT〗〗 (31)
J=150 (Tmax-Tmin)/Tmin (32)
|D-J|<10 (33)
Where, Tmax and Tmin represent the maximum and minimum temperatures in this study. [25].
*) In the conclusions it would be desirable to outline at least one line of future research.
Answer: Thanks for your observation.
Probably, as previously proposed the non-equilibrium state between phases in the VLE will be focused around the variations of vapor flow quality, entropy generation and exergy variation [26]. And probably the analysis of the phase equilibrium in magnetorheological fluids will be a second research opportunity as non-equilibrium systems [27 -28].
*) The problem of the thermodynamics of mixtures is of great interest as evidenced by the innumerable applications, also in the industrial field, that this research topic involves.
Answer: Thanks for your suggestion . It was included in the introduction.
On the other, the problem of the thermodynamics of mixtures is of great interest as evi-denced by the innumerable applications, also in the industrial field, that this research topic involves. In mixtures where azeotropic systems are present, activity coefficients data is of great utility for the design of efficient separation and purification processes [11]. The objective of this work is based on the evaluation of activity coefficients and the mathematical modeling of vapor-liquid equilibrium (VLE) binary mixtures of THF/Organic Mixtures from literature data using activity coefficient models like local composition models (Wilson and NRTL), local distribution model (UNIQUAC) and UNIFAC activity coefficients.
ANSWER: ACCEPTED. THANKS FOR YOUR COMMENT.
*) In particular, some types of mixtures require that the problem is not framed in the context of classical thermodynamics but rather in the extended irreversible thermodynamics. This is the case of magnetorheological fluids and, in particular, of certain types of evolutionary models of fludi mixture. Without prejudice to the fact that this topic is beyond the objectives of the work presented by the authors, I recommend inserting at least one sentence in the text that highlights this problem by putting the following relevant works in the bibliography:
ANSWER: ACCEPTED. THANKS FOR YOUR COMMENT.
And probably the analysis of the phase equilibrium in magnetorheological fluids will be a second research opportunity as non-equilibrium systems [27 -28].
Probably, the next future of the VLE analysis will consider the non-equilibrium state between the phases as already proposed around the variations of vapor flow quality, entropy generation and exergy variation. (as proposed in.doi:10.1016/j.ijmultiphaseflow.2012.02.008)
doi: 10.1016/j.ijnonlinmec.2019.103288
doi: 10.1007/s001610200085
Answer: the references were included.

Reviewer 2 Report
The manuscript "Study of thermodynamic modelling of isothermal and isobaric binary mixtures in vapor-liquid equilibrium (VLE) of Tetrahydrofuran with Benzene (303.15K) cyclohexane (338.15K) and methanol (103kPa)" reportsthe approximation data for some binary systems contained THF.The work is quite short and looks like a very brief discussion of well-known models. Thus, there is no novelty in this work.
I found the paper may be accepted for publication in Entropy journal after major revisions that are described below:
- A more detailed justification of the relevance and fundamentally new data or conclusions should be added to the article.
- Introduction: the choice of the systems under consideration should be presented. Also, why do the authors take into account only one temperature (or pressure) for one system? The justification of the choice must be presented in the Introduction.
- Conclusion: The authors claim “The isobaric VLE data were measured experimentally”, but these are not the data obtained in this work.
- Figures 1-3: Check, please, X-axis signature – “x,y” or “X,Y”. Also, there are many types of lines in the same plot. It is difficult to understand and compare the data. In Fig.3 some lines have sharp curvatures. How the authors can discuss this fact?
- Hole the text: check, please, carefully the spaces before references. There are spaces in some places, but not in others. Compare, for example, “global THF_[3] market, in the midst of the COVID-19 crisis[4].”
- Page 1, Lines 34-35: The reference 3 mention would be better to place after the word “market”, not after “THF”
- Page 1, Line 43: “Vapor-Liquid Equilibria” but not “ELV”.
- Title: “Modeling”, but not “modelling" (double-l)
Author Response
The manuscript "Study of thermodynamic modelling of isothermal and isobaric binary mixtures in vapor-liquid equilibrium (VLE) of Tetrahydrofuran with Benzene (303.15K) cyclohexane (338.15K) and methanol (103kPa)" reportsthe approximation data for some binary systems contained THF.The work is quite short and looks like a very brief discussion of well-known models. Thus, there is no novelty in this work.
Answer: On the other, the problem of the thermodynamics of mixtures is of great interest as evi-denced by the innumerable applications, also in the industrial field, that this research topic involves. In mixtures where azeotropic systems are present, activity coefficients data is of great utility for the design of efficient separation and purification processes [11]. The objective of this work is based on the evaluation of activity coefficients and the mathematical modeling of vapor-liquid equilibrium (VLE) binary mixtures of THF/Organic Mixtures from literature data using activity coefficient models like local composition models (Wilson and NRTL), local distribution model (UNIQUAC) and UNIFAC activity coefficients.
I found the paper may be accepted for publication in Entropy journal after major revisions that are described below:
- A more detailed justification of the relevance and fundamentally new data or conclusions should be added to the article.
Answer: The experimental isothermal VLE data 12, 13 and the predicted for the binary system of THF(1) + benzene (2), and THF(1) + cyclohexane(2) at 303.15 and 338.15 K respectively and isobaric VLE Data 14, 15 and the predicted THF(1) + methanol(2) and THF(1) + ethanol(2) at 103 kPa were corralated by Wilson, NRTL UNIQUAC and UNIFAC models. The best thermodynamic consistency the Herington test for the VLE data was found for THF(1) +cyclohexane (2) isothermal system and THF(1) + ethanol(2) isobaric system. The RMSDPi and RMSDyi show that the UNIQUAC model for isothermal systems THF(1) + benzene(2) and THF(1) + cyclohexane(2) performs superior than the other models. NRTL model for the isobaric system THF(1) + methanol(2) and UNIQUAC THF(1) + ethanol(2) perform better than the other models. Probably, as previously proposed the non-equilibrium state between phases in the VLE will be focused around the variations of vapor flow quality, entropy generation and exergy variation [26]. And probably the analysis of the phase equilibrium in magnetorheological fluids will be a second research opportunity as non-equilibrium systems [27 -28].
THF(1) + ethanol(2) isobaric system was added.
- Introduction: the choice of the systems under consideration should be presented. Also, why do the authors take into account only one temperature (or pressure) for one system? The justification of the choice must be presented in the Introduction.
Answer: Thanks for your observations. The introduction was modified: Even THF as a solvent is used in many useful chemical processes manufacturing activities and active additive in the synthesis of pharmaceutical products, it is also found with other solvents as cyclohexane in chemical waste liquids [2]. The mixture THF and cyclohexane presents an azeotrope at 338.74 K with the composition of THF at 93 % (wt %) with an almost impossible separation by conventional distillation [3]. On the other hand, in polymerization reactions, THF is soluble in all proportions with alcohols, phenols and all common solvents [4]. The separation of azeotropic multicomponent mixtures as THF, methanol or ethanol and water provide considerable potential for the combination of pervaporation and distillation processes for THF recovery with reliable benefits [5].
In the study of preferential interaction of polymers in mixed solvents, the binary mixtures of THF with aromatic hydrocarbons showed important changes at high THF con-centrations [6]. Besides, THF was used for the selective protonation of aromatic hydro-carbons with high selectivity in moderate to good yields [7] and in electro-reduction mechanism of aromatic hydrocarbons [8].
Four systems were compared:
- Conclusion: The authors claim “The isobaric VLE data were measured experimentally”, but these are not the data obtained in this work.
Answer: Thanks for your observation. The correction was done.
- Figures 1-3: Check, please, X-axis signature – “x,y” or “X,Y”. Also, there are many types of lines in the same plot. It is difficult to understand and compare the data. In Fig.3 some lines have sharp curvatures. How the authors can discuss this fact?
Answer: Thanks for your observation. The correction was done. The authors does not argue about the sharp curvatures which correspond to experimental data obtained by literature.
- Hole the text: check, please, carefully the spaces before references. There are spaces in some places, but not in others. Compare, for example, “global THF_[3] market, in the midst of the COVID-19 crisis[4].”
Answer: Thanks for your observation. The correction was done.
- Page 1, Lines 34-35: The reference 3 mention would be better to place after the word “market”, not after “THF”
Answer: Thanks for your observation. The correction was done.
- Page 1, Line 43: “Vapor-Liquid Equilibria” but not “ELV”.
Answer: Thanks for your observation. The correction was done.
- Title: “Modeling”, but not “modelling" (double-l)
Answer: Thanks for your observation. The correction was done.

Round 2
Reviewer 1 Report
All suggestions have been implemented. Therefore, the paper deserves publications. Congratulations!